# Improving access to care and community health in Haiti with optimized community health worker placement

**Clara Champagne**[1,2]*, **Andrew Sunil Rajkumar**[3], **Paul Auxila**[4], **Giulia Perrone**[5], **Marvin Plötz**[3], **Alyssa Young**[6,7], **Samuel Bazaz Jazayeri**[1,2], **Harriet G. Napier**[8], **Arnaud Le Menach**[8], **Katherine Battle**[9], **Punam Amratia**[10], **Ewan Cameron**[10], **Jean-Patrick Alfred**[11], **Yves-Gaston Deslouches**[11], **Emilie Pothin**[1,2,8]

**1** Department of Epidemiology and Public Health, Swiss Tropical and Public Health Institute, Basel, Switzerland, **2** University of Basel, Basel, Switzerland, **3** World Bank, Washington, DC, United States of America, **4** Global Financing Facility, Port-au-Prince, Haiti, **5** Global Fund, Geneva, Switzerland, **6** Clinton Health Access Initiative, Port-au-Prince, Haiti, **7** Department of Tropical Medicine, Tulane University School of Public Health and Tropical Medicine, New Orleans, LA, United States of America, **8** Clinton Health Access Initiative, Boston, MA, United States of America, **9** Institute for Disease Modeling, Seattle, WA, United States of America, **10** Telethon Kids Institute, Perth, Australia, **11** Ministère de la santé publique et de la population, Port-au-Prince, Haiti

* clara.champagne@swisstph.ch

**Data Availability Statement:** The data used in this work can be accessed from the following sources:
- Population distribution: https://www.ciesin.

## Abstract

The national deployment of polyvalent community health workers (CHWs) is a constitutive part of the strategy initiated by the Ministry of Health to accelerate efforts towards universal health coverage in Haiti. Its implementation requires the planning of future recruitment and deployment activities for which mathematical modelling tools can provide useful support by exploring optimised placement scenarios based on access to care and population distribution. We combined existing gridded estimates of population and travel times with optimisation methods to derive theoretical CHW geographical placement scenarios including constraints on walking time and the number of people served per CHW. Four national-scale scenarios that align with total numbers of existing CHWs and that ensure that the walking time for each CHW does not exceed a predefined threshold are compared. The first scenario accounts for population distribution in rural and urban areas only, while the other three also incorporate in different ways the proximity of existing health centres. Comparing these scenarios to the current distribution, insufficient number of CHWs is systematically identified in several departments and gaps in access to health care are identified within all departments. These results highlight current suboptimal distribution of CHWs and emphasize the need to consider an optimal (re-)allocation.

## Introduction

Due to social and political vulnerabilities, as well as a number of natural disasters, the current health system in Haiti cannot guarantee accessible and quality health services to the majority

columbia.edu/data/hrsl/ - Friction surface: https://malariaatlas.org/research-project/accessibility-to-cities/ - Positioning of health facilities, SPA survey, Haiti: https://www.dhsprogram.com/methodology/survey/survey-display-532.cfm The R code used (R version 4.0.3) is available at https://github.com/SwissTPH/CHWplacement.

**Funding:** This work was supported by the Global Fund (CC&EP - IQC No: 20178525) and the Bill & Melinda Gates Foundation (CC&EP - INV-030449). Under the grant conditions of the Foundation, a Creative Commons Attribution 4.0 Generic License has already been assigned to the Author Accepted Manuscript version that might arise from this submission. The funders had no role in study design, data collection and analysis, decision to publish, or preparation of the manuscript.

**Competing interests:** The authors have declared that no competing interests exist.

of the population [1, 2]. It was estimated that in 2013 only 23% of the Haitian population had access to good quality primary care, a proportion dropping to 5% in rural areas [3]. In order to achieve national objectives around universal health coverage, multifactorial challenges have been highlighted [4] and community health worker (CHW) programs have been identified as key platforms for organising and delivering interventions that can improve health access in underserved areas and reduce health access inequities [2]. These programs are also increasingly recognized by many countries as cost-effective platforms [5, 6] for improving reach, equity and quality of care [7] with high potential for national scale [8]. In Haiti, CHWs have been deployed for decades, contributing to decisive advances in tuberculosis and HIV care [9–11]. However, a significant proportion of the CHWs currently working in the country are managed by disconnected vertical programs with fixed timelines and a national survey on the CHW network in 2019 identified that only 27% of the population could benefit from a complete CHW coverage [12, 13]. Therefore, a strategy prioritizing well-coordinated and sustainable approaches for the whole community health network could enhance its impact nationally and across disease areas. Specifically, the deployment of polyvalent CHWs covering populations in urban, rural and difficult-to-reach areas is a central part of the person-centred primary care reinforcement initiated by the Ministry of Health (MoH) and will accelerate efforts to reach universal health coverage. Planning for the future recruitment and deployment of CHWs at a central level is a critical component in designing the national strategy [13]. However, forecasting needs for such locally tailored programs can be a complex task, due to numerous local constraints, such as effective accessibility and effective health facility coverage, which are important factors of physical healthcare access but are difficult to measure and incorporate when deriving a national strategy. For example, the CHWs catchment areas may be very large where population density is low, forcing the CHWs to travel long distances to access the population they have been assigned to [14]. Such constraints also need to be considered in light of the finite financial resources available for these programs [15]. In this context, mathematical modelling tools can help synthesise the available information, and explore optimal placement scenarios for CHWs over the whole Haitian territory, accounting for constraints regarding existing health access coverage, terrain and population distribution.

Geographical placement of CHWs has already been modelled using different methods that could be categorised into two approaches. On the one hand, similar location-allocation problems, have been frequently formulated as constrained optimisation problems, such as p-median or covering-based models, with various assumptions and solving methods [16, 17]. Such methodologies based on linear programming have often been used for health facility placement but also to inform CHW allocation strategies. In particular, in Haiti, Parker et al. used linear programming methods, not to study CHW geographical allocation, but to optimize how much time and resources CHWs should dedicate to different tasks in order to maximize health benefit [9]. In Malawi, Kunkel et al. implemented p-median models to optimize CHW geographical allocation as well as the location of their commodity resupply in nearby health facilities [18]. Compared to p-median models, covering-based models [17] are well suited for universal coverage objectives, as they include a maximum distance constraint and hence exclude the possibility that a small number of individuals would need to travel a long distance in the optimal solution [16]. For example, the formulation by Current and Storbeck [19], building on Toregas et al. [20], calculates the minimum number of facilities (or CHWs) required to guarantee full coverage of the targeted population under the constraints that both the distance and the total number of people assigned to a facility remain below predefined thresholds. On the other hand, newly emerging sources of geospatial information are increasingly used to inform healthcare planning. In particular, several research initiatives publish maps that reconstruct population density at high resolution, using statistical methods

combining official estimates or census data with satellite images or topographical data [21]. Estimates of the time needed to travel from one location to another are also available, using data on topography, infrastructure and transportation networks [22–24]. Such subnational level estimates have been used to measure and account for context-specific factors, when designing optimisation placement models for CHWs. For example, Toh et al. used an algorithm relying on such high-resolution surfaces to identify the optimal locations of community health services in northern Ghana, although without including maximum capacity constraints [25]. Oliphant et al. also used high resolution mapping to suggest CHW scale-up scenarios [26], using another placement methodology [22] that relies on sequential allocation of CHWs rather than constrained optimization which provides an overall optimal solution.

In this work, we combined existing fine-graded estimates of population [27] and travel times [23, 24] with implementations of Current and Storbeck's constrained optimization program [19] to inform the geographical deployment of CHWs in Haiti. In order to give guidance on important operational limitations, parameters account for population density in rural and urban areas, as well as proximity to existing health facilities, and guarantee constraints on walking time and number of inhabitants for which each CHW is responsible. Several national-scale scenarios adapted to the Haitian context are compared, in order to inform the number and distribution of CHWs required to bridge the gap in access to health services. The results of the analysis advised the development of the national Strategic Community Health Plan [28] by providing guidance on the number and geographical distribution of CHWs.

## Results

Four scenarios of CHWs spatial optimization accounting for population distribution and travel time were selected within an iterative process of interactions between modellers and decision makers to correspond as much as possible to the specific needs of the Haitian community health programme (Fig 1). All scenarios ensure that the walking time between the CHW position and their allocated households does not exceed 60 minutes, and that the number of inhabitants allocated to each CHW remains below predefined thresholds that vary across scenarios. In scenario A, the entire territory is covered by CHWs, with varying population thresholds (i.e., maximal number of people per CHW) in rural, urban and metropolitan areas (where rural and urban areas are defined based on population density, cf. Material and Methods section and S1 Fig). Scenario B focuses exclusively on difficult-to-reach areas, such that only households situated far from a community health centre (*Centre communautaire de santé*, CCS) may be covered by CHWs, assuming that these centres can take over some of the CHW activities in the areas not covered by CHWs. Finally, scenarios C and C2 provide a compromise between the previous two scenarios: the entire territory is covered by CHWs but the maximum population thresholds depend not only on the rural or urban status, but on the distance to the nearest CCS, such that areas far from any CCS are prioritized. In order to derive the parameters for these four scenarios, several threshold values for maximal population, and distance to CCSs aligning with official guidelines and desired CHW workload were explored, and among these, the ones leading to total CHW numbers in the same order of magnitude as the current one (i.e. approximately 4000 CHWs nationally) were selected.

The results for each scenario are presented for the Grande-Anse department in Fig 2 (other departments are presented in S1 Text, and in S1 File), with associated population, friction and accessibility of CCS maps for interpretability. For example, in the Grande-Anse department, all scenarios except scenario B recommend to position many CHWs in densely populated areas, with sometimes up to 5 CHWs recommended in the same 1km$^2$ location (cf. Fig 2 panels

## Model assumptions

**CHW maximum walking time**

≤ 60'

**Maximum population visited**

≤ **max**

**Walking time to closest Comunity Health Center (CCS)**

close ≤ t

far > t

close to CCS

far from CSS

## Simulated scenarios

### Scenario A

**Non-Metropolitan**

Rural 1000    Urban 2500

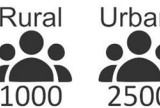

**Metropolitan**

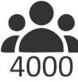
4000

### Scenario B

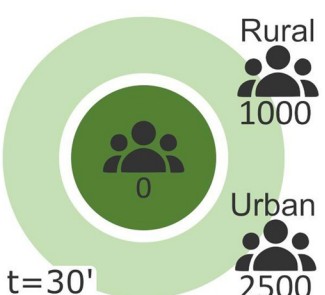

Rural 1000

Urban 2500

t=30'

### Scenario C

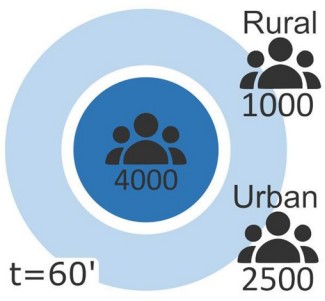

Rural 1000

4000

Urban 2500

t=60'

### Scenario C2

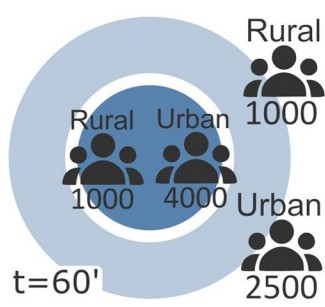

Rural 1000

Rural 1000  Urban 4000

Urban 2500

t=60'

**Fig 1. Description of the model assumptions in the four scenarios.** The maximum walking time is fixed to 60 minutes. The maximum number of inhabitants per CHW is varied across the four scenarios. In scenario A, the entire territory is covered by CHWs, with a maximum population of 1000 per CHW in rural areas, 2500 in urban areas and 4000 in the metropolitan area. In Scenario B, only areas situated at more than a 30 minutes' walk from a community health centre (CCS) are covered by CHWs, with a maximum population of 2500 per CHW in urban and metropolitan areas and 1000 in rural areas. In scenarios C and C2, the entire territory is covered by CHWs but the maximum population thresholds depend on the distance to the nearest CCS. In scenario C, less than a 60 minutes' walk from a CCS, 4000 people are assigned to each CHW and more than a 60 minutes' walk from a CCS, the maximum populations is 2500 per CHW in urban and metropolitan areas and 1000 in rural areas. Scenario C2 is similar to scenario C, except that the maximum population is 1000 in rural areas, whatever the distance to the closest CCS, and the 4000 threshold within a 60 minutes' walk from a CCS is applied only for urban areas.

1 and 2), but also disperse at least one CHW in difficult-to-reach areas of the south of the department (cf. Fig 2 panels 1 and 3).

The exact total number of CHWs required at the national level under the different scenarios is presented in Fig 3 (panel 1). As expected by design, the four scenarios lead to total numbers of CHWs between 3800 and 5200, which is close to the current number (3698 CHWs according to the 2017–2018 Service Provision Assessment survey (SPA) [31], 4337 active CHWs according to the 2019 CHW mapping [13]), indicating that the chosen constraints on maximal population and walking time might correspond to realistic thresholds when designing a national strategy with similar CHW total numbers. Nevertheless, the four scenarios also differ in several aspects.

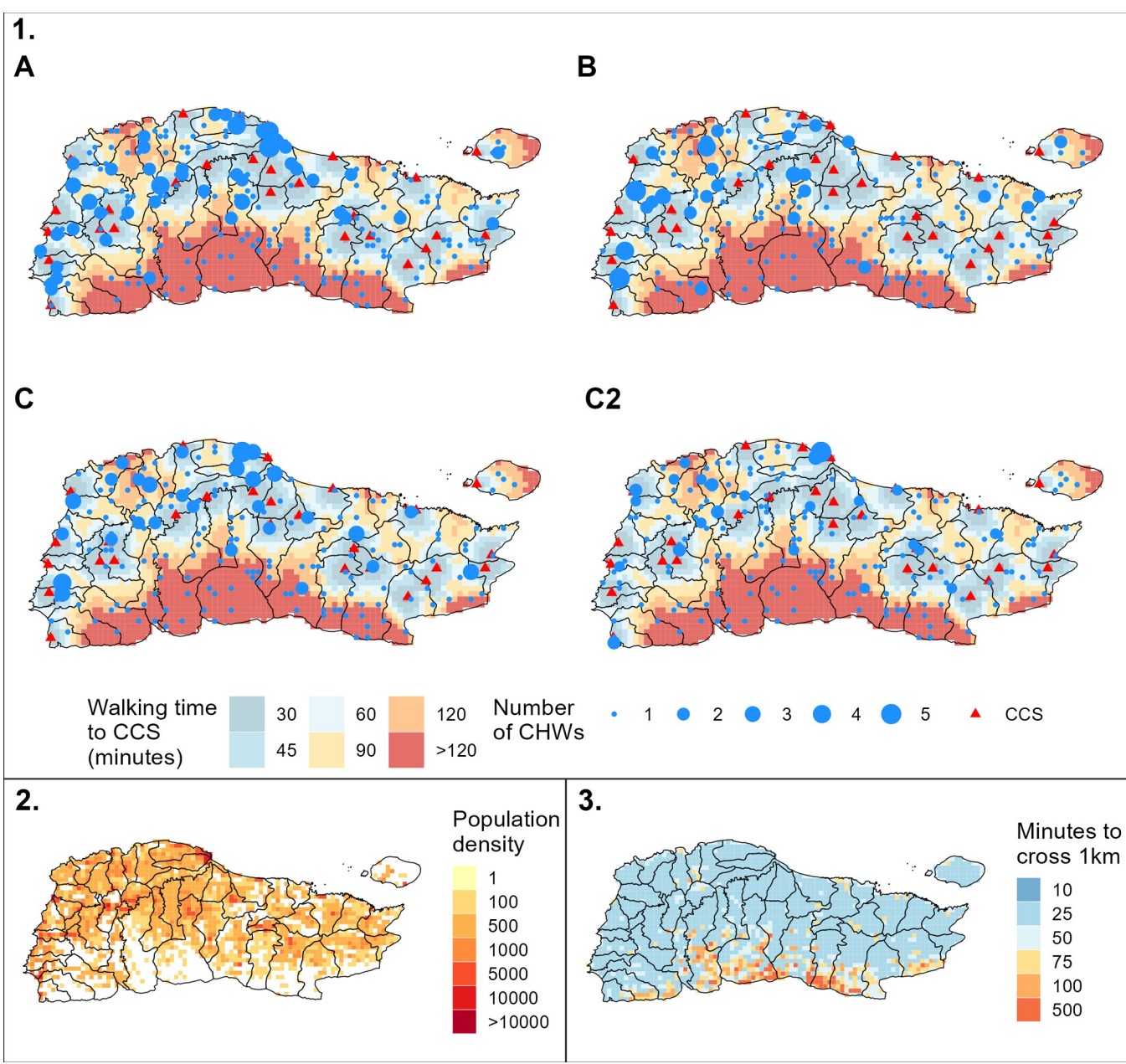

**Fig 2. Comparison of the placement scenarios in the Grande-Anse department.** 1. The four CHW placement scenarios (A, B, C and C2). CHW positions are indicated with blue dots, community health centres (CCS) are indicated with red triangles. The colored surface indicates the predicted walking time to the closest CCS using the methodology by Weiss and colleagues [23, 24]: difficult-to-reach areas, located more than 60 minutes walk from the nearest CCS, are shown in orange/red and areas with easier access (less than 60 minutes walk) are shown in blue. 2. For interpretability: prediction of population density in 2020 per square kilometre [27, 29]. 3. For interpretability: walking time friction surface by Weiss et al. is shown as the time required to cross 1km [23]. The shapefile from the Centre National de l'Information Géo-Spatiale (CNIGS) was used [30] (available at https://data.humdata.org/dataset/hti-polbndl-adm1-cnigs-zip).

The total number of required CHWs is lowest in scenario B, since in this scenario, the CHWs are exclusively focused on underserved households located more than 30 minutes from a community health center, as opposed to the other scenarios where CHWs are positioned in the whole territory. Among these other three scenarios, scenario A, which relies only on population density for prioritisation, leads to the highest number of CHWs because it assigns on

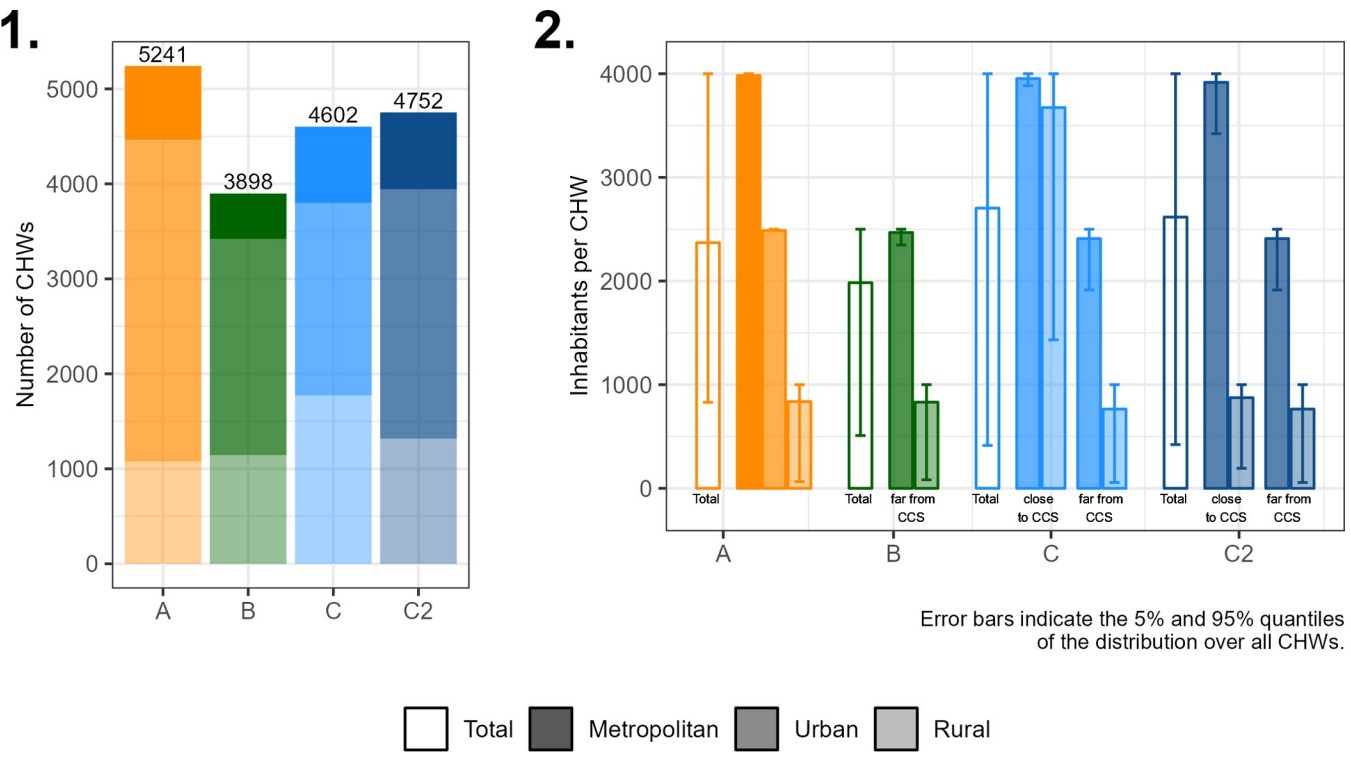

**Fig 3. Comparison of the placement scenarios at the national level.** 1. Total number of CHW required in each scenario and proportion of CHWs affected to urban, rural and metropolitan areas. 2. Actual average number of inhabitants assigned per CHW and per scenario; the error bars represent the 5% and 95% quantiles of the distribution over all CHWs. In this panel, the metropolitan area is only defined for scenario A; in scenario B, the travel time defining areas close to a community health centre (CCS) is 30 minutes; in scenarios C and C2, it is 60 minutes (cf. Fig 1).

average a lower number of inhabitants per CHW (except in the metropolitan area). The last two scenarios, C and C2, in which the number of CHWs is reduced in the vicinity of CCSs, lead to intermediate total numbers. These two scenarios, in particular scenario C, also lead to the highest proportion of CHWs in rural areas due to the combined effect of the different criteria used. In scenario C2, more CHWs are required than in scenario C, as the number of inhabitants per CHW is constrained to be lower in rural areas. However, fewer CHWs are positioned in rural areas in scenario C2 than in C, because some CHWs placed in urban areas may be assigned inhabitants from the neighbouring rural areas.

As the defined maximum number of people that could be assigned per CHW in each scenario (Fig 1) is an indicative threshold that is not always reached, Fig 3 panel 2 indicates the actual average number of people that would be assigned to each CHW for each of the scenarios, as well as the 5 and 95% quantiles of the distribution over all the CHWs. For example, in scenario C, although the maximum threshold is set at 4,000 in areas close to a CCS, the average number of people assigned per CHW is about 2700 nationally. Across the scenarios, between 40 and 60% of the CHWs are assigned the maximum number of inhabitants allowed in their category (S2 Fig). Additionally, in all scenarios, a small number of CHWs are assigned to very few inhabitants: in particular, less than 2.5% of CHWs are assigned fewer than 100 people (S2 Fig). This happens mainly in very difficult-to-reach places, where the 60-minute walking time condition cannot be satisfied easily. In other situations, the algorithms suggest the placement of several CHWs within a small densely populated area with varying population size in their catchment areas, and the potential unequal workload could be reallocated between neighbouring CHWs, when defining the effective catchment area for each CHW individually.

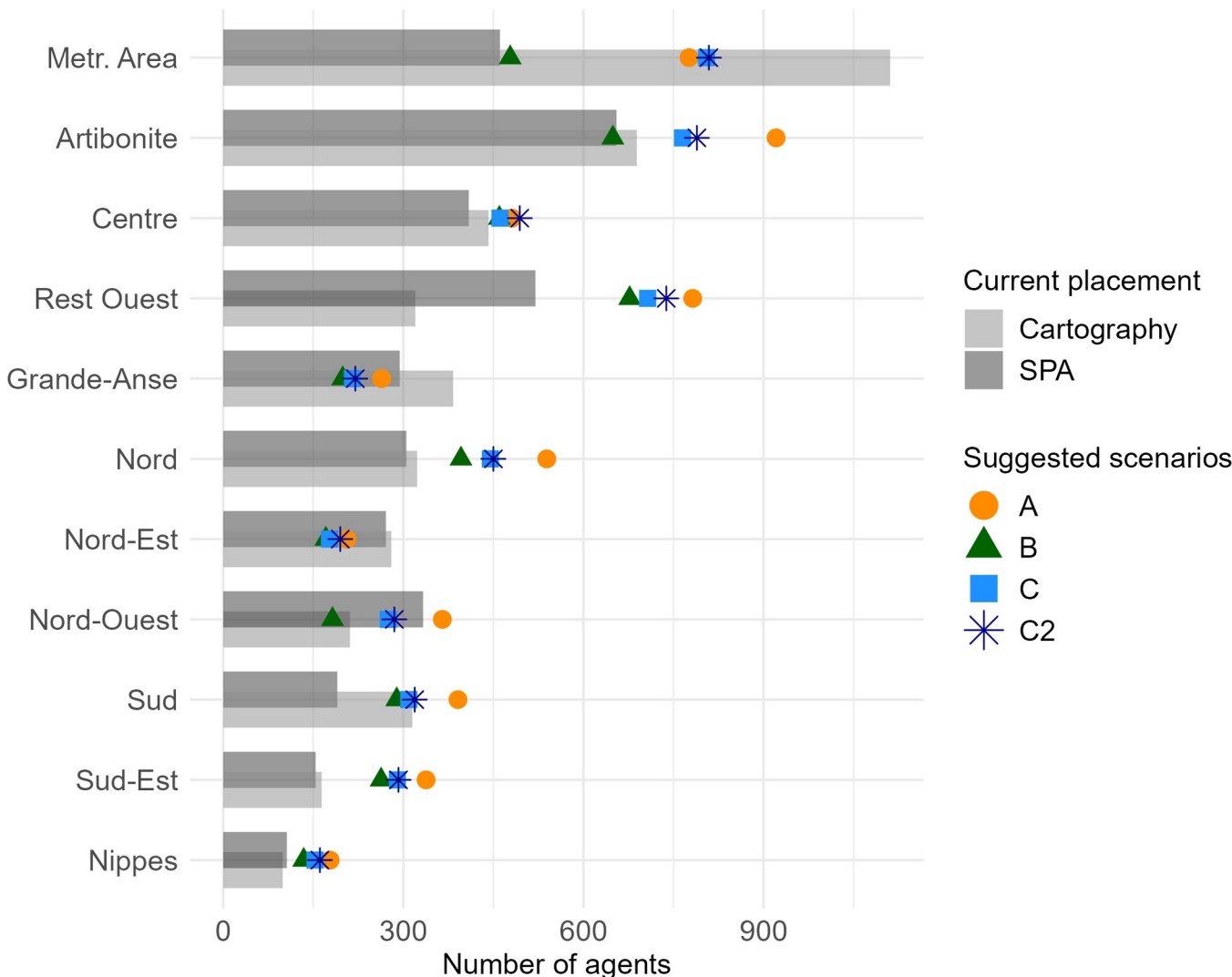

**Fig 4. Comparison by department of the current number of CHWs according to the SPA survey [31] and the CHW mapping ("Cartography", [13]), and the number of CHWs suggested in scenarios A, B, C and C2.**

The comparison by department between the current number of CHWs and the number of CHWs suggested in the four scenarios is presented in Fig 4. Despite the differences from one scenario to another, the departments of Sud-Est, Ouest (excluding the Metropolitan Area), Nord, and to a lesser extent Centre and Nippes are systematically identified as having an insufficient number of CHWs. On the other hand, the Nord-Est and Grande-Anse departments have a sufficient total number of CHWs according to the four scenarios.

In order to compare the suggested scenarios and the current placement at a finer spatial scale, the distribution of the CHWs by section communale is compared to the CHWs totals from the SPA survey (Fig 5) with scenario C as an example, as this scenario was favoured by the MoH (the other scenarios are displayed in S3 Fig). Despite the findings at the departmental level (Fig 4), areas with a CHW deficit are identified within all departments (see areas shown in red in Fig 5 panel 2). Thus, even in the Nord-Est department, for which the overall number of CHWs is indicated as sufficient, some sections communales would require an increase in the number of CHWs.

**1.**

**2.**

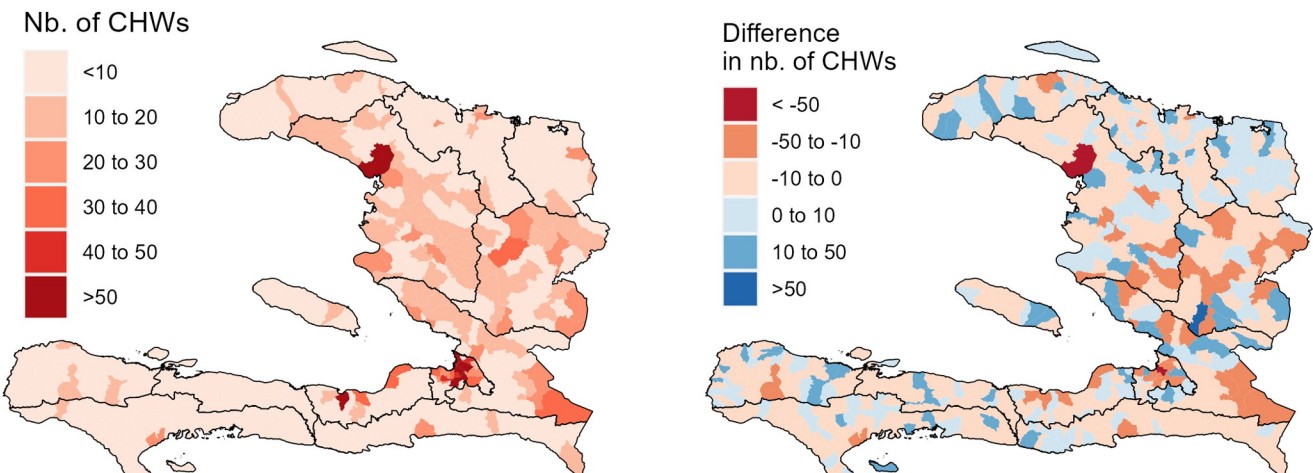

**Fig 5. Analysis of scenario C, where CHWs are positioned in the whole territory, accounting for the position of CCSs.** 1. Number of CHWs per section communale, according to scenario C. 2. Difference by section communale between the current number of CHWs in the SPA survey and the suggested number of CHWs under the scenario C. Negative values (signified in red) indicate a deficit, positive values (in blue) signify a surplus. The shapefile from the Centre National de l'Information Géo-Spatiale (CNIGS) was used [30] (available at https://data.humdata.org/dataset/hti-polbndl-adm1-cnigs-zip).

Some robustness checks are performed on scenario C by modifying independently the scenario parameters by increasing or decreasing them by 10%. The effects at the national level are presented in S4 and S5 Figs and in S1 File. As expected, the total number of required CHWs is influenced by the parameter values: it is larger with more conservative constraints (reduced walking time, reduced CHW:population ratio, reduced distance to health facilities, increased population defined as rural), and lower when constraints are relaxed (increased walking time, increased CHW:population ratio, increased distance to health facilities, decreased population defined as rural). The most influential parameters are clearly the maximum number of people for which a given CHW is responsible (capacity) and to a lesser extent, the maximum walking time. In particular, under scenario C, reducing the CHW capacity by only 10% would lead to an almost 10% increase in the total number of CHWs required (i.e., more than 400 additional CHWs nationally). However, the relative share of CHWs in urban and rural areas and the effective number of inhabitants per CHW do not vary strongly in the range of parameter values tested. Additionally, the effects at the departmental level are presented in S6 Fig: notably, the variations between departments are of much larger magnitude than the variations due to changes in parameters.

## Discussion

In this work, four scenarios for CHW geographical placement in Haiti were designed, accounting for population density, topography and the location of existing health facilities, in concertation with the Ministry of Health. First, these scenarios combine predefined constraints on workload for the CHWs in terms of walking time and assigned population to define different CHW catchment areas, while requiring a total national number of CHWs that is close to the current one. Second, these national theoretical scenarios were compared with coverage of the existing CHW network, to identify regions where there was a current gap in CHW coverage,

namely the departments of Sud-Est, Ouest and Nord, and to a lesser extent Centre and Sud. These findings corroborate those of the national CHW mapping conducted by the Ministry of Health [13]. At a lower geographical level (section communale), gaps in access to CHWs were identified within all departments, even when the total number of CHWs at the departmental level is indicated as sufficient to cover the needs. These results highlight high spatial heterogeneity in access to health services within a department, and therefore emphasize the need for an optimal (re-) allocation of CHWs that is driven by locally validated data. The results of the analysis advised the development of the national Strategic Community Health Plan [28] and the suggested number of CHWs per section communale was provided to the program to guide future microplanning work.

With this methodology, geographical placement scenarios are derived from pre-defined assumptions on CHW catchment areas in terms of travel time and population ratios. On one hand, in order to ensure that CHWs do not have to walk long distances to perform their duties, walking time for a CHW to reach a household was constrained below 60 min, a threshold mentioned or used in other analyses [14, 26]. On the other hand, the total number of people assigned per CHW was constrained to remain below a threshold based on assumptions on CHW workload. The highest threshold considered (4000) was high compared to other countries [32] and above what is specified within the national guidelines (up to 2500 in urban areas), and associated workload calculations relied on the assumption of a 40 hour work week. However, lower capacity values would correspond to a significant increase in the total national CHW numbers (S4–S6 Figs). The compromise found was to restrict this high population threshold to the metropolitan area (scenario A) or to the vicinity of community health centers (scenarios C and C2). The other areas thus have lower capacity thresholds and are given higher priority. Among the four scenarios, the Ministry of Health favoured scenario C, in which CHWs are positioned in communities both within and beyond 60 minutes travel time from the nearest health centre at a standard ratio of 1 CHW:4000 population within the 60 minute radius and 1:000 and 1:2500 beyond the radius in rural and urban areas respectively [28]. This scenario affords access to CHWs in communities near and far from a health centre, with heightened coverage in communities identified as difficult-to-reach, regardless of the metropolitan area status.

As most of the input data is publicly available (population surfaces, friction surfaces), this analysis could be applied to other contexts. The model is flexible for country-specific use and can be adapted to the large differences across countries regarding the role and status of CHWs [32, 33]. In the present analysis, distance to existing health facilities was accounted for in order to prioritise areas where access to care is clearly reduced by physical distance. In other situations, disease-specific risk could also be accounted for. For instance, in order to prioritise areas where significant malaria transmission is identified, the placement algorithm can be modified to include malaria incidence into the model in addition to capacity and walking time constraints, either by giving more weight or lower population numbers per CHW in malaria endemic areas, or by restricting the allocation to the areas where malaria risk is considered present with a predefined probability (exceedance probability maps) [34]. Additionally, detailed workload data can be used to calibrate the maximum capacity per CHW, which is very specific to the characteristics of a given CHW program [32]. Regarding travel time estimates, the predictions for walking time were used for the purpose of this analysis, but in other contexts other friction maps including motorized travel or additional country-specific features may be more appropriate [23, 26]. Finally, this methodology was applied at country scale to support the elaboration of a national plan, but it could also be adapted to be used at finer spatial scales to help during implementation by local stakeholders.

This analysis, however, has several limitations. First and foremost, the results obtained are conditioned by the data used. In particular, the guarantee of total population coverage depends on the reliability of the estimates used to represent population totals, distribution and travel times on a small scale. For example, the model ensures that a CHW will not have to walk more than one hour to reach each of their assigned households, provided that the estimation of travel times and walking speed as well as the prediction of household locations are accurate, and that the CHW lives exactly in the 1km$^2$ area recommended by the model. Additionally, the distinction between rural and urban areas was defined based on modelled population density, and may not correspond to the exact delineations of official administrative areas. These correspond to average assumptions, therefore these results are mainly valid when aggregated at higher levels and/or if more accurate input data is provided (e.g. geolocation of all households). Nevertheless, even though these high-resolution maps need some careful scrutiny on their strengths and weaknesses before being used, they represent an opportunity to use modelling tools for informing planning decisions.

Secondly, operational considerations and practical challenges [35] may lead to deviations from the theoretical plan. The aim of this modelling work is to provide a guidance tool for the recruitment and deployment of CHWs, which will need to be adapted locally to account for additional practical constraints. For example, in very low-density areas, where the model predicts a very low number of inhabitants per CHW, specific knowledge of the local context is required to disentangle real healthcare needs from uncertainties linked to the geospatial representation of population distribution and travel time. In other situations, where the algorithm suggests the placement of several CHWs within a small densely populated area, the effective catchment area for each CHW would need to be defined individually. The question of how to supervise, support and provide commodities to CHWs in areas situated very far from existing health facilities would also need to be addressed. With the model, it is possible to allocate each CHW to the closest health facility or health centre, but the validity of such fine scale predictions would need to be verified for each CHW specifically. Additionally, CHWs cannot be recruited, relocated or redeployed easily, due to employment contracts, requirements that CHWs belong to the community to which they are assigned, and the importance of a community-informed selection process [13]. These aspects are beyond the scope of the current analysis, and a micro-planning analysis is therefore needed to translate the suggested scenarios into an operational deployment plan.

Thirdly, the method relies on the assumptions made. The analysis focused exclusively on physical accessibility and did not consider other determinants of access to care, such as financial accessibility [4], the quality of services provided [36], and more generally demographic, economic and social factors impeding or facilitating healthcare access [37, 38]. This analysis also focused exclusively on population coverage assuming that CHWs serve all individuals equally and therefore did not consider how the decisions regarding the geographical distribution of CHWs could depend on the presence of epidemiological risks, for example higher malaria risk in the Grande-Anse and Sud departments [34], or particular needs in specific population subgroups. Furthermore, the choice of thresholds for the maximum distance and maximum population in urban and rural areas could lead to border effects and gives a tendency for the model to place more CHWs in high-capacity areas, in order to minimise their total number. However, defining thresholds rather than optimising the average distance to care has the advantage to guarantee access to care within a certain distance to all individuals, in line with the universal access criteria. Additionally, robustness checks were performed to assess the sensitivity of the results to these thresholds. Overall, this methodology makes it possible to explore different thresholds and evaluate how they translate to CHW total numbers and distribution per department.

Finally, this calculation requires complex solvers as the computational burden increases with the size of the geographical area to be analysed. Although this may reduce the ease of use of the model, it has the advantage of guaranteeing optimisation of the solutions, contrary to heuristic methods [17]. Other heuristic approaches also exist, such as the algorithm implemented in the AccessMod software, where new facilities (or CHWs) are added sequentially in areas where the population without access is highest or based on other prioritisation criteria [22, 26, 39]. With our approach, the locations of all CHWs are optimised simultaneously, which, in certain circumstances, may lead to lower total CHWs numbers required to reach universal coverage. Additionally, in order to facilitate the use of our results for routine decision making, the methodology is available as an R package to interface integer programming solvers and an interactive visualization is provided in S1 File.

Despite these limitations, this methodology to simulate optimal placement scenarios for CHWs can be a helpful thinking tool for programmatic decision making related to community health programmes while accounting for constraints on travel time, population density and access to care. It can also help to identify coverage gaps and priority areas for scale-up according to pre-defined objectives for universal coverage or burden reduction.

In conclusion, geospatial information and optimisation methods were combined to investigate scenarios for geographic placement of community health workers in Haiti. This work has highlighted high heterogeneity in CHW access in the country and has informed the national Strategic Community Health Plan of the Haitian Ministry of Health [28]. It could be readily applied in other contexts to support the planning of evidence-based community health strategies.

## Material and methods

### Data and inputs

**High-resolution geospatial estimates.** In order to capture population spread at fine geographical resolution, the High Resolution Settlement Layer [27, 40, 41] was used. This map is based on the use of satellite images to reconstruct the location of buildings, and then distribute the population into areas of about 30m$^2$. This approach was selected among other high resolution population maps [21] because its methodology allowed for a more realistic representation of population distribution in low density areas, which is of critical importance when assessing access to care. A similar choice was made in [26]. In order to ensure that the total population estimates align with the 2020 projections from the Haitian Institute of Statistics and Informatics (IHSI) [29], this map was aggregated to 1km$^2$ resolution and then raked to match the 2020 population projects in each section communale [42]. Additional data from the IHSI exist at the enumeration section level, but they are not georeferenced and therefore not available for the current analysis.

Based on this high-resolution population map, so-called urban and rural areas were defined following the European Commission's "degree of urbanization approach" described in the 2017 World Bank report [43], i.e. cells with a population density higher than 300 people per km$^2$ are grouped into clusters of contiguous cells, and the clusters with more than 2,000 people are defined as "urban clusters" [43]. With this methodology, at the national level there are about 9.8 million people living in urban areas and 2.7 million in rural areas (cf. S1 Fig and S2 Text), with rural areas representing more than 75% of the inhabited territory. The communes of Port-au-Prince, Delmas, Cité Soleil, Tabarre, Carrefour and Pétion-Ville constitute the metropolitan area, and represent a total of 3.1 million inhabitants. Overall, with these definitions, 25% of the Haitian population live in the metropolitan area, 54% live in non-metropolitan urban areas and 21% live in non-metropolitan rural areas.

In order to quantify travel times, estimates of a friction map that predicts the walking time required to travel from one point to another in the country and produced by the Malaria Atlas Project [23, 24] were used. The use of walking time rather than distance per kilometre allows for differences in topography to be taken into account, making this measure for access to health care services more realistic.

**Surveys on health services in Haiti.** The Service Provision Assessment (SPA) survey on health services in Haiti [31] was used to identify the location of health facilities. This assessment, conducted in 2017–2018, identified 1033 structures present in the country and was able to survey 1007; GPS location is available for 1019 of them, of which 355 are community health centres (*centres communautaires de santé*, CCS). Among health facilities, CCSs are the structures where CHWs are preferentially attached to and they provide community health services. By combining these data with the friction map [23, 24, 44], it is possible to identify difficult-to-reach areas, located more than 60 minutes by foot from any CCS, assuming that individuals living in the vicinity of a CCS have an easier access to community health services. With these data, it is estimated that 32% of the population lives at more than 60 minutes by foot from any CCS (and 62% lives at more than 30 minutes, cf. S2 Text).

The SPA survey also provides the number of CHWs–polyvalent agents, i.e. agents de santé communautaires polyvalents (ASCP) and vertical agents, i.e agents de santé communautaires (ASC)–assigned to each health facility. At the national level, this survey indicates a total of 3698 CHWs in the country in 2017–2018. Additionally, a detailed mapping of the CHWs employed and working in Haiti was conducted by the Ministry of Health in 2019 [13]. This survey identified 4337 active agents ("agents intégrés") at the national level. The departmental-level total estimates from this survey were used in complement with the SPA survey and compared to results from simulated scenarios (A, B, C and C2 in Fig 1).

## Algorithms for CHW placement

**Hypotheses and country specific scenarios.** The method for CHW placement relies on two main hypotheses (Fig 1): (1) the CHWs should not have to walk more than a certain time to reach the households in their catchment area (maximum walking time), and (2) the number of people assigned to a CHW should not exceed a certain threshold (maximum number of people per CHW).

In this work, the maximum walking time is fixed to 60 minutes, indicating that all households are situated at less than one hour (by foot) from their CHW, as it was highlighted that 2 hours round trip was a desirable walking distance for CHWs [14].

Regarding the maximal number of people per CHW, the MoH recommendation is 1 CHW per 2500 inhabitants in urban areas and 1 CHW per 1000 inhabitants in rural areas [13]. Considering average values with the definition of urban and rural areas used here and ignoring the effects of uneven population spatial distribution, this would require more than 6500 CHWs in the country, which is significantly higher than the current resources. Therefore, in this work, threshold values based on the expected workload of the CHWs were explored with the following assumptions:

1. A CHW can visit 8 households per day (45 minutes per visit and the to- and from- 60 minutes walking time back to CHW site correspond to 8 working hours, assuming a 40 hour work week).

2. The average size of a household is 4.3 people [45].

3. A year has 220 working days.

As a result, multiple thresholds for the maximum number of people per CHW were selected based on assumptions on frequency of visits per year that might vary by settings. In particular, thresholds of 1000, 2500 and 4000 inhabitants per CHW were assumed to correspond respectively to 8, 3 and 2 visits per household per year on average (S1 Table), the first two thresholds thus being aligned with the MoH recommendation. These assumptions are based on average values and therefore do not represent the exact CHW's daily planning but rather aim to ensure that the CHW's workload remains reasonable. In addition, the maximum number of people is a threshold that is not necessarily reached: for example, with an assumption of 1000 inhabitants per CHW, it is possible that a CHW may be assigned less than 1000 people in very low density areas, so that their walking time always remains below 60 minutes per household.

Additional features can be included. For instance, if CHWs are deployed specifically to provide health services in difficult-to-reach areas, the analysis can be restricted to areas situated more than a certain distance away from health facilities. Another possibility is to allow for higher maximum population numbers in the surroundings of existing health facilities, assuming that services available at the facilities overlap with tasks of the CHWs.

In order to derive scenarios adapted to the Haitian context, the maximum number of people per CHW was varied depending on urbanization level and proximity to CCSs–all scenarios ensuring that the distance between CHWs and households remain below 60 minutes. We explored several combinations of the constraints on maximal population (1000, 2500, 4000 inhabitants per CHW), and distance to CCSs (0, 30, 60 minutes). Among these combinations, we selected the ones leading to a total numbers of CHWs in the same order of magnitude as the current number according to the 2017–2018 SPA survey [31] and the CHW mapping [13], i.e. approximately 4000. This led to the four scenarios presented in Fig 1.

For comparison purposes, the total numbers of CHWs required if only the population thresholds were accounted for, i.e., without considering travel time and population geographical distribution were also calculated. These totals are presented and commented in S2 Text.

**Linear programming and location covering problems.** In order to determine the geolocation of the CHWs, the CSCLP optimization programme (Capacitated Set Covering Location Problem, by Current and Storbeck [19]) is used. This method calculates an optimal location for all the CHWs simultaneously, including both constraints (maximum walking time and maximum number of people per CHW). More precisely, it searches for the minimum number of CHWs required to reach universal coverage under the walking time and population constraints. The version by Current and Storbeck [19] assumes that there cannot be several facilities (here CHWs) placed on the same position because potential facility sites are treated as binary variables. This resulted in infeasible solutions in densely populated areas. We modified the algorithm to allow for several CHWs per sites, by treating potential CHW sites as positive integer variables, without any upper bound.

The CSCLP optimization algorithm by Current and Storbeck [19] considers:

$$\min Z = \sum_{j \in J} Y_j$$

Subject to

$\sum_{j \in N_i} X_{ij} \geq 1 \ \forall i \in I$ (universal coverage constraint), with $N_i = \{j \in J | d_{ij} \leq S\}$ (maximum walking time constraint)

$\sum_{i \in I} a_i X_{ij} - k_j Y_j \leq 0 \ \forall j \in J$ (capacity constraint)

$Y_j \in \mathbb{N} \ \forall j \in J$ (the numbers of CHWs are integers) where $Y_j$ is the number of CHWs positioned in location j, $a_i$ is the number of inhabitants living in location i, $X_{ij}$ is the proportion of inhabitants from location i assigned to a CHW in location j and $k_j$ is the maximum number of

people that a CHW placed in location j could cover (capacity). $d_{ij}$ indicates the walking time between location i and j calculated using the friction surface [23], and S is the maximum walking time per CHW. Additionally, J consists of the potential CHW locations and I are the locations where inhabitants that need to be covered are situated. The parameter values in each of the four scenarios corresponding to Fig 1 are indicated in S2 Table.

Calculations were performed using the Gurobi commercial solver version 8.1 [46] and the lpsolveAPI R package [47, 48], which retrieves the exact solution of the optimization problem. Although heuristic methods exist, it was chosen, as in Kunkel et al. [18], to use an exact solver with a controlled tolerance parameter (called "Gurobi gap" in [18]). Additionally, the optimization program was run for each department independently (with Artibonite, Centre and Reste-Ouest each sub-divided into two), in order to reduce computational burden, but also because the operational decision-making is done at the department level. The "Gurobi gap" was set between 5 and 6 CHWs per department, which was considered to be negligible compared to the current national number of CHWs (for the 10 departments: 60 << 4000).

To enhance the transferability of this methodology to other contexts, the code is available in the form of an R package that permits the design of placement scenarios using any input maps on population and friction surfaces. It can be coupled with external optimisation solvers to be used in practice with real data with large maps and complex allocation problems. The current implementation provided in the package can be coupled with Gurobi [46] or SCIP [49].

In order to explore the sensitivity of the results to the threshold values selected, some robustness checks are performed on scenario C. Independently, the thresholds used for the maximum walking time, the distance to the closest CCS that define difficult-to-reach areas, the maximum population and the definition of urban areas (minimum density and minimum population) are increased or decreased by 10% (S4–S6 Figs). Therefore, the thresholds used for the maximum walking time and the distance to the closest CCS that define difficult-to-reach areas are varied to 54min and 66 min). The maximum population thresholds are varied to 2250 and 2750 in urban areas, 900 and 1000 in rural areas and 3600 and 4400 in the vicinity of health centers. The thresholds used in the definition of urban areas were also varied: the minimum density was varied to 270 or 330, and the minimum population was varied to 1800 and 2200. These changes can be visualized dynamically in S1 File.

## Supporting information

**S1 Fig. Urban and rural areas defined in the model.** Calculated following the European Commission's "degree of urbanization approach" described in [43] based on predictions of population density in 2020 per square kilometre [27,29]. The shapefile from the Centre National de l'Information Géo-Spatiale (CNIGS) was used [30] (available at https://data.humdata.org/dataset/hti-polbndl-adm1-cnigs-zip).
(TIFF)

**S2 Fig. Additional results on the total number of CHWs per scenario.** 1. Percentage of CHWs in each scenario who are assigned to the threshold number of inhabitants as defined in Fig 1 and S2 Table. 2. Percentage of CHWs in each scenario who are assigned to less than 100 inhabitants. The metropolitan area is only defined for scenario A; in scenario B, the travel time defining areas close to a community health centre (CCS) is 30 minutes; in scenarios C and C2, it is 60 minutes (cf. Fig 1).
(TIFF)

**S3 Fig. Comparison of the scenarios and the current placement per sections communales for scenarios A, B and C2.** 1. Number of CHWs per section communale, according to

scenarios A, B or C2 respectively. 2. Difference by section communale between the current number of CHWs in the SPA survey and the suggested number of CHWs under the scenarios A, B or C2 respectively. Negative values (signified in red) indicate a deficit, positive values (in blue) signify a surplus. The shapefile from the Centre National de l'Information Géo-Spatiale (CNIGS) was used [30] (available at https://data.humdata.org/dataset/hti-polbndl-adm1-cnigs-zip).
(TIFF)

**S4 Fig. Total number of CHWs in scenario C when model parameters are increased or decreased by 10%.** 1. Required number of CHW at the national level. 2. Percent increase or decrease in national number of CHWs required if parameter values are increased or decreased by 10%.
(TIFF)

**S5 Fig. Average number of inhabitants per CHW in scenario C, and 5% and 95% quantiles of the distribution, when model parameters are increased or decreased by 10%.**
(TIFF)

**S6 Fig. Required number of CHW per department in scenario C, when model parameters are increased or decreased by 10%.**
(TIFF)

**S1 Table. Assumed correspondence between the number of visits per year and the threshold on the number of inhabitants assigned per CHW.**
(PDF)

**S2 Table. Description of the four scenarios selected.** CCS refers to community health centres (*centres communautaires de santé*). *L* denotes the set of locations containing a CCS, *J* the potential CHW locations. Additional notations are provided in the main text.
(PDF)

**S1 Text. Presentation of the four scenarios per department.**
(PDF)

**S2 Text. Hypothetical total CHW numbers if travel time and population distribution were not considered.**
(PDF)

**S1 File. Interactive visualisation of the CHW placement scenarios at the national level.**
(HTML)

## Acknowledgments

This work was presented during the workshop entitled "Atelier d'élaboration du Plan Stratégique Santé Communautaire" (February 17th-21st 2020) and the present analysis takes into account the comments received during this workshop. The authors also acknowledge Tatiana Alonso Amor for her support in the visualisation, especially for Fig 1, Munir Winkel for suggestions on the visualisation of Fig 4, Katya Galactionova for helpful discussions and Laura Skrip for helpful comments on the manuscript. Calculations were performed at sciCORE (http://scicore.unibas.ch/) scientific computing center at University of Basel.

## Author Contributions

**Conceptualization:** Clara Champagne, Andrew Sunil Rajkumar, Paul Auxila, Giulia Perrone, Arnaud Le Menach, Jean-Patrick Alfred, Yves-Gaston Deslouches, Emilie Pothin.

**Data curation:** Clara Champagne, Alyssa Young, Katherine Battle, Punam Amratia.

**Formal analysis:** Clara Champagne, Katherine Battle, Punam Amratia.

**Funding acquisition:** Emilie Pothin.

**Methodology:** Clara Champagne, Andrew Sunil Rajkumar, Paul Auxila, Giulia Perrone, Marvin Plötz, Emilie Pothin.

**Software:** Clara Champagne, Samuel Bazaz Jazayeri, Emilie Pothin.

**Supervision:** Emilie Pothin.

**Visualization:** Samuel Bazaz Jazayeri.

**Writing – original draft:** Clara Champagne.

**Writing – review & editing:** Clara Champagne, Andrew Sunil Rajkumar, Paul Auxila, Giulia Perrone, Marvin Plötz, Alyssa Young, Harriet G. Napier, Arnaud Le Menach, Katherine Battle, Punam Amratia, Ewan Cameron, Jean-Patrick Alfred, Yves-Gaston Deslouches, Emilie Pothin.

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
