## [Decision Letter · Decision Letter 0]

2 Nov 2021

PGPH-D-21-00700

Improving access to care and community health in Haiti with optimized community health worker placement.

Dear Dr. Champagne,

Thank you for submitting your manuscript to PLOS Global Public Health. After careful consideration, we feel that it has merit but does not fully meet PLOS Global Public Health’s publication criteria as it currently stands. Therefore, we invite you to submit a revised version of the manuscript that addresses the points raised during the review process.

The manuscript has been assessed by our reviewers. They have raised a minor point which we believe would improve the manuscript and may allow a revised version to be published in PLOS Global Public Health.

We look forward to receiving your revised manuscript.

Kind regards,

Jitendra Kumar Singh, PhD

Academic Editor

Journal Requirements:

1. Please update the completed 'Competing Interests' statement, including any COIs declared by your co-authors. If you have no competing interests to declare, please state "The authors have declared that no competing interests exist". Otherwise please declare all competing interests beginning with the statement "I have read the journal's policy and the authors of this manuscript have the following competing interests:"

2. Please amend your detailed Financial Disclosure statement. This is published with the article, therefore should be completed in full sentences and contain the exact wording you wish to be published.

i). State the initials, alongside each funding source, of each author to receive each grant.

ii). State what role the funders took in the study. If the funders had no role in your study, please state: “The funders had no role in study design, data collection and analysis, decision to publish, or preparation of the manuscript.”

Reviewers' comments:

Reviewer's Responses to Questions

**Comments to the Author**

1. Does this manuscript meet PLOS Global Public Health’s publication criteria? Is the manuscript technically sound, and do the data support the conclusions? The manuscript must describe methodologically and ethically rigorous research with conclusions that are appropriately drawn based on the data presented.

Reviewer #1: Yes

Reviewer #2: Yes

2. Has the statistical analysis been performed appropriately and rigorously?

Reviewer #1: Yes

Reviewer #2: Yes

3. Have the authors made all data underlying the findings in their manuscript fully available (please refer to the Data Availability Statement at the start of the manuscript PDF file)?

Reviewer #1: Yes

Reviewer #2: Yes

4. Is the manuscript presented in an intelligible fashion and written in standard English?

Reviewer #1: Yes

Reviewer #2: Yes

5. Review Comments to the Author

Reviewer #1: This is an excellent piece of modelling based work, executed nicely. Work itself merits publication. However, I am afraid, in actual practice, this is never going to be useful in program setting, due to reasons know to every public health professional. I congratulate the team for coming up with such modelling work. Atleast it may inspire others to apply in some other settings which may be of use.

Reviewer #2: Ensure the citations of authors is accurate; the authorship of a local Haitian author was unclear and the absence of this contributing author would seriously impinge on the credibility of the article, not to mention the ethics of global health research done to, and not with, local stakeholder engagement and ownership.

6. PLOS authors have the option to publish the peer review history of their article (what does this mean?). If published, this will include your full peer review and any attached files.

**Do you want your identity to be public for this peer review?** For information about this choice, including consent withdrawal, please see our Privacy Policy.

Reviewer #1: **Yes: **Dr Arun Kumar Aggarwal

Reviewer #2: No

---

## [Editor Report · Decision Letter 1]

9 Mar 2022

Improving access to care and community health in Haiti with optimized community health worker placement.

PGPH-D-21-00700R1

Dear Dr. Champagne,

We are pleased to inform you that your manuscript 'Improving access to care and community health in Haiti with optimized community health worker placement.' has been provisionally accepted for publication in PLOS Global Public Health.

Best regards,

Jitendra Kumar Singh, PhD

Academic Editor